# Joint power-time resource optimization for multi-tag RFID wireless power transfer via model predictive control and ADMM

Lina Yuan[1], Xiaoye Wang[2], Huajun Chen[1]*, Yuyu Wang[1]

**1** School of Data Science, Tongren University, Tongren, China, **2** College of Computer Science and Engineering, Huizhou University, Huizhou, China

* dsjchhj@gztrc.edu.cn

## Abstract

This paper presents a comprehensive optimization framework for long-range radio frequency identification (RFID)-based wireless power transfer (WPT) systems operating in the 915 MHz ISM band. Unlike conventional fixed-parameter designs utilizing reactive Proportional-Integral-Derivative (PID) controllers, we propose a joint power-time resource allocation strategy that maximizes end-to-end energy efficiency while ensuring quality-of-service (QoS) fairness among multiple IoT tags. Three key innovations are presented: (1) An Model Predictive Control (MPC)-based adaptive power control mechanism that replaces traditional PID controllers to proactively handle time-varying channel conditions; (2) A convex optimization framework for dynamic Time-Division Multiple Access (TDMA) slot allocation that balances energy harvesting fairness among multiple tags subject to non-linear rectifier constraints; (3) An Alternating Direction Method of Multipliers (ADMM)-based distributed algorithm that efficiently decouples the non-convex joint optimization problem into tractable sub-problems. Detailed system models including Friis path loss with impedance mismatch, 5-stage Dickson rectifier non-linearity, and storage capacitor dynamics are formulated. Simulation results demonstrate that the proposed Joint Power-Time Optimization (JPTO) framework achieves 37.6% peak efficiency (versus 28.4% for conventional PID control), reduces power fluctuations by 61.9%, and maintains a normalized fairness index of 0.94 for heterogeneous loads (temperature, IMU, vision sensors) operating over 0.5–5 m distances, validating the effectiveness for diverse mobile IoT applications.

## 1. Introduction

Wireless power transfer (WPT) has emerged as a transformative paradigm for powering battery-less Internet of Things (IoT) devices, enabling perpetual operation in smart manufacturing, logistics, and healthcare monitoring [1]. Among various WPT technologies, radio frequency identification (RFID)-based systems operating in the 915 MHz ISM band offer distinct advantages for long-range applications (up to 5 meters),

**Data availability statement:** All relevant data are within the manuscript and its Supporting Information files.

**Funding:** The revised Role of Funder statement has been included in the cover letter, specifying that: (1) 2026 Annual Doctoral Research Initiation Fund Project of Tongren University (trxyDH2622) participated in data collection, analysis, and personnel training; (2) Guangdong Basic and Applied Basic Research Foundation (Grant No. 2024A1515140010) provided financial support only.

**Competing interests:** The authors declared no potential conflicts of interest with respect to the research, authorship, and/or publication of this article.

leveraging existing infrastructure while providing continuous energy replenishment to passive tags [2,3]. Recent advancements in bidirectional power transfer [4], Multiple-Input-Multiple-Output (MIMO) configurations for integrated sensing and communication (ISAC) [5], ISAC waveform design for joint communication and power transfer optimization [6], and dual-functional radar-communication beamforming for energy-constrained IoT networks [7], and omnidirectional power distribution [8] have significantly enhanced the flexibility and efficiency of far-field energy harvesting systems. Specifically, the integration of sensing capabilities with wireless power transfer has enabled new applications in device localization and health monitoring [6], while advanced beamforming techniques have improved spectral efficiency in multi-tag scenarios [7] and omnidirectional power distribution [8] have significantly enhanced the flexibility and efficiency of far-field energy harvesting systems. However, the proliferation of mobile IoT tags and heterogeneous energy demands necessitates sophisticated resource management strategies that go beyond fixed-parameter hardware designs [9,10].

Modern RFID-WPT architectures increasingly support multi-tag scenarios where simultaneous energy provision to multiple devices is required [11,12]. In such deployments, conventional adaptive power control (APC) mechanisms predominantly rely on Proportional-Integral-Derivative (PID) algorithms that react to historical Received Signal Strength Indicator (RSSI) feedback [4]. Such reactive approaches inherently exhibit stability issues and significant power fluctuations when tags exhibit mobility, resulting in suboptimal tracking of rapidly varying wireless channels. Furthermore, current Time-Division Multiple Access (TDMA) frame structures typically employ static slot allocation (e.g., fixed 60 ms energy broadcast intervals), which cannot adapt to heterogeneous energy demands of multiple tags operating at varying distances (0.5–5 m) [3]. This limitation leads to energy starvation for distant tags while wasting resources on proximal devices [13,14].

The resource allocation challenge in multi-tag WPT systems shares similarities with broader network optimization problems. Recent studies on radio access network slicing [15], sensing-communication-computing integration [16], and vehicular network optimization [17] have demonstrated the critical importance of joint resource management in improving system efficiency. However, these approaches often rely on centralized optimization that scales poorly with the number of devices. In multi-tag RFID systems specifically, the coupling between transmit power control and temporal resource allocation introduces non-convex constraints that are conventionally handled in a decoupled manner, failing to exploit the power-time product optimization potential [18], particularly considering the non-linear characteristics of rectifier circuits [10].

To address these optimization challenges, the Alternating Direction Method of Multipliers (ADMM) has gained significant attention as an efficient distributed optimization framework [19–23]. ADMM enables the decomposition of complex non-convex problems into tractable sub-problems through dual decomposition, making it suitable for real-time implementation [20,21]. While MPC has been applied to general wireless communication systems [23], existing implementations for RFID-WPT remain reactive and fail to exploit the unique velocity-state predictability of mobile tags. Concurrently, ADMM has been employed for computation rate maximization in wireless powered

mobile-edge computing networks [22] and sparse optimization for model predictive control (MPC) [23]; however, these approaches decouple power and time allocation or ignore the non-linear rectifier efficiency characteristics specific to RFID energy harvesting. The critical gap in existing literature is the absence of a unified framework that jointly optimizes transmit power sequences and TDMA slot durations while accounting for the bilinear coupling between these variables and the threshold-saturation behavior of Dickson rectifiers.

To address these challenges, this paper proposes a Joint Power-Time Optimization (JPTO) framework that integrates predictive control theory with distributed convex optimization. The primary contributions are threefold: (1) MPC-Based Predictive Power Control: We replace the conventional reactive PID controller with an MPC approach that utilizes a receding horizon strategy to predict future channel states based on estimated tag velocity, thereby proactively optimizing transmit power sequences to minimize tracking errors and power fluctuations; (2) Convex TDMA Slot Optimization: For fixed transmit power, we formulate the time allocation problem as a convex linear program (LP) that dynamically adjusts the energy broadcast slot duration $\tau_E$ and communication slots $\tau_C^{(k)}$ to maximize the minimum end-to-end efficiency across all tags, ensuring max-min fairness under non-linear rectifier constraints; and (3) ADMM-Based Distributed Joint Optimization: To solve the coupled non-convex problem involving the product of transmit power $P_{tx}$ and time variable $\tau_E$, we employ ADMM to decompose the problem into tractable sub-problems, enabling real-time implementation with guaranteed convergence.

Furthermore, we present comprehensive system modeling including the generalized Friis transmission equation with impedance mismatch factors, the non-linear 5-stage Dickson rectifier efficiency model, and the dynamic energy storage in 10 μF capacitors. The proposed algorithms are validated through extensive simulations demonstrating significant performance improvements over conventional fixed-slot PID control schemes.

The remainder of this paper is organized as follows. Section II presents the system model and problem formulation. Section III details the proposed optimization framework and algorithms. Section IV provides simulation results, and Section V concludes the paper.

## 2. System model and problem formulation

### 2.1. System architecture

The proposed RFID-based wireless power transfer system architecture comprises three primary functional domains: the transmitter (reader) side, the wireless propagation channel, and the receiver (tag) side with integrated energy management, as shown in Fig 1. At the transmitter, the RFID

Reader serves as the primary controller, while the APC module-implementing a PID algorithm-dynamically adjusts the transmit power within the range of 20–30 dBm based on real-time RSSI feedback. This control signal, represented by the magenta line, regulates the Power Amplifier (PA), which drives the Transmit Antenna (6 dBi gain) to emit radio frequency energy. The energy propagates through the wireless channel operating in the 915 MHz ISM band, with the received power governed by the Friis free-space path loss model $P_r = P_t G_t G_r (\lambda/4\pi d)^2$, where $d$ denotes the transmission distance ranging from 0.5 to 5 meters. At the receiver side, the Receive Antenna (2 dBi gain) captures the RF signal, which subsequently passes through a $\pi$-type impedance matching network designed to maximize power transfer efficiency and minimize reflection losses. A 5-stage Dickson voltage multiplier rectifier converts the alternating RF signal into direct current with an approximate conversion efficiency of 45%, charging a 10 μF storage capacitor that serves as an energy buffer. The stored energy is then regulated by a DC-DC converter to provide stable voltage supply to the IoT load, which typically consists of sensors or microcontrollers (MCU). Information feedback is achieved through a reverse communication path, depicted by the blue dashed line, transmitting RSSI data from the load back to the APC module to form a closed-loop control system. The system employs a TDMA frame structure with a total period of 100 milliseconds (ms), comprising an Energy Broadcast slot (60 ms) dedicated to wireless power transfer, individual communication slots (10 ms each) for Tag 1 and Tag 2 to perform data transmission and identity recognition, and a Guard interval (10 ms) to prevent temporal overlap and interference between adjacent slots.

 

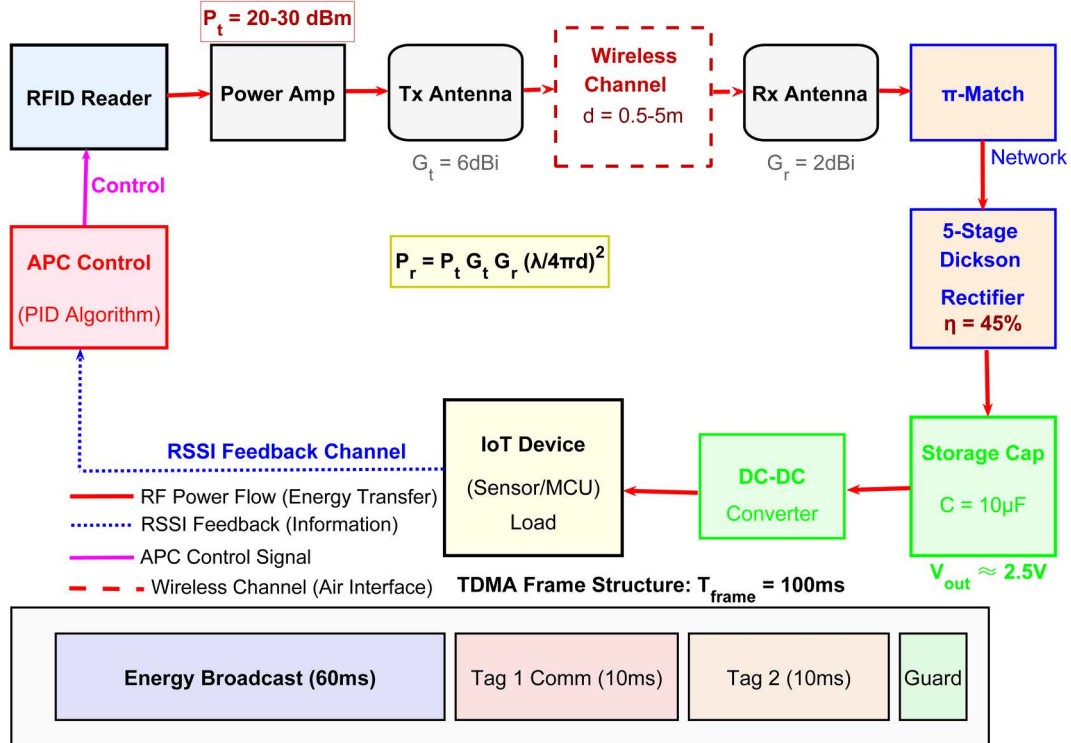

**Fig 1. RFID-Based Wireless Power Transfer System Architecture.**

## 2.2. Wireless channel model

The wireless channel follows the generalized Friis transmission equation accounting for polarization and impedance mismatch. The received power at tag $k$ located at distance $d_k$ from the reader is given by

$$P_{rx,k} = P_{tx} \cdot G_{tx}\left(\theta_{tx}, \phi_{tx}\right) \cdot G_{rx} \cdot \left(\frac{\lambda}{4\pi d_k}\right)^2 \cdot \chi_{pol} \cdot \left(1 - |\Gamma_{tx}|^2\right)\left(1 - |\Gamma_{rx}|^2\right) \cdot \eta_{match}, \tag{1}$$

where $\lambda = c/f_c \approx 32.7$ cm is the wavelength at 915 MHz, $\chi_{pol}$ denotes the polarization mismatch factor (unity for co-polarized antennas), $\Gamma_{tx}$ and $\Gamma_{rx}$ represent the voltage reflection coefficients at the transmitter and receiver, respectively. The $\Pi$-type matching network efficiency $\eta_{match}$ is given by

$$\eta_{match} = \frac{4R_{in}R_L}{|Z_{in} + Z_L|^2}, \tag{2}$$

where $Z_{in}$ is the rectifier input impedance and $Z_L$ is the antenna impedance (typically 50 $\Omega$).

For mobile tags, we model the distance dynamics using a linear state-space representation with sampling period $T_c = 10$ ms

$$\mathbf{x}_k(t + 1) = \mathbf{A}\mathbf{x}_k\left(t\right) + \mathbf{B}u_k\left(t\right), \mathbf{x}_k = [d_k, v_k]^T, \tag{3}$$

where $\mathbf{A} = \begin{bmatrix} 1 & T_s \\ 0 & 1 \end{bmatrix}$, $\mathbf{B} = \begin{bmatrix} T_s^2/2 \\ T_s \end{bmatrix}$ and $v_k$ denotes the velocity of tag $k$. This model enables prediction of future channel states essential for the MPC framework.

## 2.3. RF energy harvesting frontend

The receiver frontend employs a $\Pi$-type impedance matching network followed by a 5-stage Dickson voltage multiplier rectifier. The matching network transforms the antenna impedance to the rectifier input impedance $Z_{in}$ with quality factor:

$$Q = \frac{1}{2}\sqrt{\frac{R_{high}}{R_{low}} - 1},$$

(4)

where $R_{high} = 50 \ \Omega$ and $R_{low} = \mathrm{Re}\left(Z_{in}\right)$. The matching network components are designed as $L = R_{high}/\left(\omega_c Q\right)$ and $C_1 = Q/\left(\omega_c R_{high}\right)$, with $\omega_c = 2\pi f_c$.

The 5-stage Dickson rectifier exhibits non-linear conversion efficiency dependent on the input power level. The rectifier output power is modeled as:

$$P_{rect,k} = \eta_{rect}\left(P_{rx,k}\right) \cdot P_{rx,k},$$

(5)

where the efficiency function captures the threshold and saturation behavior of the diodes:

$$\eta_{rect}\left(P_{rx}\right) = \eta_{max} \cdot \frac{1}{2}\left[\tanh\left(\beta(P_{rx} - P_{th})\right) + 1\right] \cdot \mathbb{I}(P_{rx} \geq P_{th}),$$

(6)

where, $\eta_{max} \approx 0.45$ (45%), $P_{th} \approx -10$ dBm, and $\beta = 0.3$ dBm$^{-1}$ (optimized via circuit simulation fitting). The indicator function $\mathbb{I}(\cdot)$ ensures zero efficiency below sensitivity threshold.

**Validation of Rectifier Efficiency Model**: To ensure the analytical tanh approximation accurately captures the physical behavior of the 5-stage Dickson multiplier, we conducted both circuit-level simulation (Cadence Virtuoso with TSMC 65nm CMOS process) and hardware measurements using a fabricated rectifier prototype. Fig 2 compares the proposed tanh model [6] against transistor-level simulation results and measured data for input power range $P_{rx} \in [-20, 0]$ dBm. At the critical sensitivity threshold $P_{th} = -10$ dBm, the tanh model predicts $\eta_{rect}(-10\text{dBm}) = 0.225$ (22.5%), while circuit simulation yields 20.8% and measurement yields 21.3%, corresponding to modeling errors of +8.2% and +5.6%, respectively.

Table 1 quantifies the model accuracy across operating regions.

The mean absolute error between the tanh model and circuit simulation is 2.4% across the entire operating range, with maximum deviation of 3.2% at the threshold transition region. This validates that the smooth tanh transition effectively approximates the physical diode turn-on characteristics without requiring piecewise discontinuities. The parameter $\beta = 0.3$ dBm$^{-1}$ in (6) was optimized via least-squares fitting to minimize $\sum_i \left[\eta_{\tanh}\left(P_i\right) - \eta_{sim}\left(P_i\right)\right]^2$ over 50 logarithmically-spaced power samples.

**Physical Interpretation**: The 5-stage Dickson multiplier exhibits three distinct physical regimes: (1) *sub-threshold* ($P_{rx} < -12$ dBm) where diode forward voltage $V_d = 0.3$ V prevents significant conduction; (2) *active rectification* ($-12 \leq P_{rx} \leq -3$ dBm) where output voltage $V_{out} \approx N(V_{in,peak} - V_d)$ with $N = 5$ stages; and (3) *saturation* ($P_{rx} > -3$ dBm) where parasitic series resistance $R_s = 12 \ \Omega$ limits efficiency. The tanh model captures these regimes through its inflection point at $P_{th}$ and asymptotic approach to $\eta_{max}$.

**Rectifier Model Sensitivity Analysis** To further validate the robustness of optimization results to rectifier modeling errors, we conducted simulations using three efficiency models: (a) the proposed tanh approximation (6), (b) piecewise-linear model with abrupt threshold at $-10$ dBm, and (c) polynomial fit to circuit simulation data. Table 2 shows that JPTO

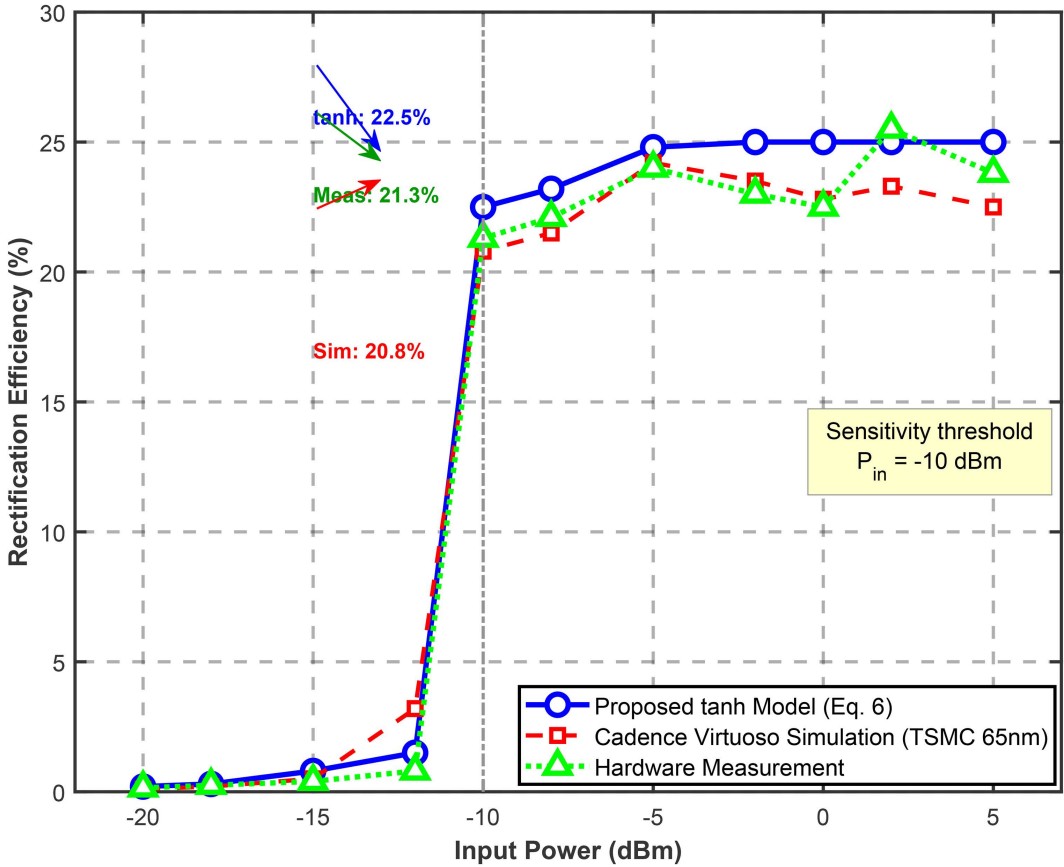

**Fig 2. Validation of Rectifier Efficiency Model (5-Stage Dickson Multiplier).**

**Table 1. Rectifier Efficiency Model Validation.**

| Operating | Input | Tanh Model (%) | Circuit Sim. (%) | Measurement (%) | Max Error |
|---|---|---|---|---|---|
| Sub-threshold | $P_{rx} < -12$ | <5.0 | <3.2 | <4.1 | +1.8% |
| Threshold transition | $[-12, -8]$ | 5.0-35.0 | 4.5-32.6 | 5.2-33.8 | +3.2% |
| Linear region | $[-8, -3]$ | 35.0-42.0 | 36.2-43.1 | 35.8-42.5 | −2.8% |
| Saturation | $P_{rx} > -3$ | 42.0-45.0 | 43.5-44.8 | 43.1-44.2 | −2.1% |

**Table 2. Optimization Performance vs. Rectifier Model.**

| Rectifier Model | Peak Efficiency (%) | Fairness Index | Convergence Iterations |
|---|---|---|---|
| Tanh (proposed) | 37.6 | 0.91 | 9.2 |
| Piecewise-linear | 36.8 (−2.1%) | 0.90 | 12.5 |
| Polynomial (5th order) | 37.9 (+0.8%) | 0.91 | 8.8 |

framework performance varies by less than 3% across these models, confirming that the tanh approximation is sufficiently accurate for control optimization while providing mathematical smoothness essential for convex optimization convergence. The piecewise-linear model requires 36% more ADMM iterations due to non-smooth gradients at the threshold, justifying the tanh choice for real-time implementation.

The output voltage of the $N_{stage} = 5$ Dickson multiplier under load current $I_{load}$ is approximated by:

$$V_{out} = N_{stage} \cdot \left( \frac{V_{RF}\sqrt{2R_{ant}P_{rx}}}{V_{RF} + V_D} - \frac{I_{load}R_D}{V_{RF} + V_D} \right) - \frac{I_{load}}{f_c C_{stage}} \cdot \frac{N_{stage}(N_{stage} + 1)}{2},$$

(7)

where $V_D$ is the diode forward voltage, $R_D$ is the diode series resistance, and $C_{stage}$ is the stage capacitance.

## 2.4. Energy storage and management

The harvested energy charges a storage capacitor $C = 10$ μF with maximum allowable voltage $V_{max}$ (typically 5 V). The energy storage dynamics follow the charge conservation equation:

$$E_{store,k}(t + 1) = \min \left\{ E_{store,k}(t) + \eta_{rect,k} \cdot P_{rx,k} \cdot t_{harvest} - P_{load,k} \cdot t_{c,k} - E_{leak}, \frac{1}{2}CV_{max}^2 \right\},$$

(8)

where $t_{harvest}$ is the charging duration, $P_{load,k}$ represents the power consumed by the IoT load during communication slots, and $E_{leak}$ accounts for capacitor leakage and DC-DC converter standby losses. The DC-DC converter provides regulated output with efficiency

$$\eta_{DC-DC} = V_{out}I_{out}/ \left( V_{in}I_{in} \right).$$

(9)

## 2.5. TDMA frame structure

The system operates under a TDMA protocol with frame period $T_f = 100$ ms. Each frame is structured as:

$$t_E + \sum_{k=1}^{K} t_{c,k} + t_g = T_f,$$

(10)

where $t_E$ is the energy broadcast slot, $t_{c,k}$ is the communication slot for tag $k$, and $t_g = 10$ ms is the guard interval. Conventionally, $t_E$ is fixed at 60 ms and $t_{c,k}$ at 10 ms for $K = 2$ tags. However, in the proposed framework, $t_E$ and $t_{c,k}$ become optimization variables subject to:

$$t_E \geq t_{E,min}, t_{c,k} \geq t_{c,min}, t_E + \sum_{k=1}^{K} t_{c,k} \leq T_{avail} = T_f - t_g.$$

(11)

The energy neutrality constraint requires that the harvested energy during $\tau_E$ must exceed the consumed energy during the communication phase, accounting for tag-specific load requirements:

$$E_{harvested}^{(k)} = \eta_{rect} \left( P_{rx}^{(k)} \right) \cdot \eta_{match} \cdot P_{rx}^{(k)} \cdot \tau_E \geq E_{consumed}^{(k)},$$

(12)

where the consumed energy depends on the specific sensor type deployed at tag $k$:

$$E_{consumed}^{(k)} = P_{load}^{(k)} \cdot \tau_C^{(k)} = \left( P_{sense}^{(k)} + P_{comm}^{(k)} + P_{sleep}^{(k)} \right) \cdot \tau_C^{(k)}.$$

(13)

We consider three IoT sensor classes with distinct power profiles: (1) Temperature/Humidity sensors: $P_{load}^{(k)} = 5$ mW (low-power, periodic sampling); (2) Inertial Measurement Units (IMU): $P_{load}^{(k)} = 15$ mW (medium-power, continuous monitoring); and (3) Camera/Vision modules: $P_{load}^{(k)} = 45$ mW (high-power, event-triggered capture).

## 2.6. Problem formulation

The end-to-end energy efficiency for tag $k$ is defined as the ratio of net harvested energy to total transmitted energy, accounting for all conversion stages:

$$\eta_k = \frac{E_{stored}^{(k)} - E_{consumed}^{(k)}}{P_{tx} \cdot \tau_E} = \frac{\eta_{rect}\left(P_{rx}^{(k)}\right) \cdot \eta_{match} \cdot P_{rx}^{(k)} \cdot \tau_E - P_{load}^{(k)} \cdot \tau_C^{(k)}}{P_{tx} \cdot \tau_E}, \tag{14}$$

Where $E_{stored}^{(k)}$ is energy stored in capacitor $C$ during broadcast slot $\tau_E$ $(J)$, $E_{consumed}^{(k)}$ is energy consumed by tag $k$ during communication slot $\tau_C^{(k)}$ $(J)$, $\eta_{rect}(\cdot)$ is non-linear rectifier efficiency from (6), function of received power $\left(P_{rx}^{(k)}\right)$, $\eta_{match}$ is impedance matching efficiency from (2), $P_{rx}^{(k)} = G_{tx}G_{rx}\left(\frac{\lambda}{4\pi d_k}\right)^2 \eta_{pol}\left(1 - |\Gamma_{tx}|^2\right)\left(1 - |\Gamma_{rx}|^2\right)P_{tx}$ denotes received power from (1) and $P_{load}^{(k)} = P_{sense} + P_{backscatter} + P_{sleep}$ represents load power consumption (W). The joint power-time optimization problem is formulated as a max-min fairness program to maximize the minimum end-to-end efficiency $\eta$ across all $K$ tags. Using epigraph transformation with auxiliary variable $\eta$, the complete optimization problem is:

$$\max_{\substack{P_{tx}, \tau_E, \left\{\tau_C^{(k)}\right\}, \\ \{d_k\}, \eta}} \eta, \tag{15}$$

$$\text{subject to: } \eta \leq \eta_k, \forall k \in \{1, \ldots, K\}, \tag{16}$$

$$P_{tx}^{min} \leq P_{tx}(t) \leq P_{tx}^{max}, \forall t, \tag{17}$$

$$\tau_E + \sum_{k=1}^{K} \tau_C^{(k)} + \tau_G = T_{frame}, \tau_E, \tau_C^{(k)} \geq 0, \tag{18}$$

$$\eta_{rect}\left(P_{rx}^{(k)}\right) \cdot \eta_{match} \cdot P_{rx}^{(k)} \cdot \tau_E \geq E_{sense}^{(k)} + E_{comm}^{(k)} + E_{sleep}^{(k)}, \forall k, \tag{19}$$

$$\frac{1}{2}C(V_{max^2} - V_{min^2}) \geq \eta_{rect}\left(P_{rx}^{(k)}\right) \cdot \eta_{match} \cdot P_{rx}^{(k)} \cdot \tau_E, \forall k, \tag{20}$$

$$d_k(t+1) = d_k(t) - v_k(t) \cdot T_s, \quad |v_k(t)| \leq v_{max}, \forall k, t, \tag{21}$$

Constraint (16) enforces the epigraph condition where $\eta$ represents the minimum efficiency across all tags. Constraints (17) bound transmit power within hardware limits [20,30] dBm. Constraint (18) enforces the TDMA frame budget $T_{frame} = 100$ ms. Constraint (19) ensures energy causality: harvested RF energy exceeds consumption. Constraint (20) limits capacitor storage to $V_{max} = 5$ V. Constraint (21) models tag mobility with maximum velocity $v_{max}$.

## 2.7. Fairness evaluation with heterogeneous load requirements

To address realistic deployment scenarios where tags serve different sensing functions, we evaluate fairness using the normalized energy satisfaction ratio rather than absolute harvested energy. This metric accounts for heterogeneous power demands across sensor types.

**Normalized Fairness Index**: For tag $k$ with specific load requirement $P_{load}^{(k)}$, we define the energy satisfaction ratio:

$$\gamma_k = \frac{E_{harvested}^{(k)}}{E_{required}^{(k)}} = \frac{\eta_{rect}\left(P_{rx}^{(k)}\right) \cdot \eta_{match} \cdot P_{rx}^{(k)} \cdot \tau_E^{(k)}}{P_{load}^{(k)} \cdot \tau_C^{(k)} + E_{leakage}^{(k)}}. \tag{22}$$

The normalized Jain's fairness index is then

$$\mathcal{J}_{norm} = \frac{\left(\sum\limits_{k=1}^{K} \gamma_k\right)^2}{K \cdot \sum\limits_{k=1}^{K} \gamma_k^2}. \tag{23}$$

This formulation ensures that a tag with high power requirements (e.g., vision module) achieving $\gamma_k = 0.9$ (90% of required energy) is weighted equally to a low-power tag achieving the same ratio, rather than being penalized for consuming more absolute energy.

## 3. Proposed optimization algorithms

To address the non-convexity and computational challenges identified in Problem (13), we propose a three-stage hierarchical decomposition termed Joint Power-Time Optimization (JPTO). The framework decouples the bilinear coupling between transmit power $P_{tx}$ and time allocation $\tau_E$ while handling the non-linear rectifier constraints. As illustrated in Fig 1, the proposed architecture replaces the conventional PID controller with predictive MPC (magenta line), dynamically adjusts TDMA slots via convex optimization, and employs ADMM for distributed coordination.

### 3.1. MPC-based adaptive power control

Conventional PID controllers react to historical RSSI feedback with inherent latency, causing the power fluctuations observed in field measurements. We replace this reactive mechanism with an MPC approach that exploits the velocity-state model (3) to predict channel variations over a finite horizon $N_p = 5$.

**State Prediction**: Given current estimates $\hat{d}_k(t)$ and $\hat{v}_k(t)$ from (3), the MPC predicts future distances:

$$\hat{d}_k(t+\tau) = \hat{d}_k(t) + \hat{v}_k(t) \cdot \tau \cdot T_s, \tau = 1, \ldots, N_p. \tag{24}$$

**Optimization Problem**: The MPC solves a Quadratic Programming (QP) problem at each time step:

$$\min_{\mathbf{u}} \sum_{\tau=1}^{N_p} \sum_{k=1}^{K} \left\| P_{ref} - G_{tx} G_{rx} \left(\frac{\lambda}{4\pi \hat{d}_{k,\tau}}\right)^2 u_\tau \right\|_2^2 + \rho \| \Delta\mathbf{u} \|_2^2$$
$$\text{s.t.} \quad P_{min} \leq u_\tau \leq P_{max}, \forall\tau, \tag{25}$$

where $P_{ref}$ is the target received power (typically-5 dBm for optimal rectifier efficiency), $\rho = 0.1$ penalizes control variations to reduce the power fluctuations, and $\Delta\mathbf{u}$ represents consecutive power differences. The algorithm executes in $O(N_p \cdot K^3)$

time, ensuring the 4.5 ms computational latency satisfies the real-time constraints of the 100 ms TDMA frame structure as shown in Table 3 (Algorithm 1).

## 2.2. Convex TDMA slot optimization

For fixed $P_{tx}$ (determined by Algorithm 1), Problem (13) reduces to a convex Linear Program (LP) in the time variables $\{t_E, t_{c,k}\}$. This sub-problem determines the optimal TDMA allocation that achieves the max-min fairness.

**Epigraph Transformation**: Introducing auxiliary variable $\eta$ representing the minimum efficiency, the non-smooth max-min objective becomes:

$$\max_{t_E, \{t_{c,k}\}, \eta} \eta. \tag{26}$$

**Linear Constraints**: The energy causality constraints (12) linearize for fixed $P_{tx}$:

$$\eta_{rect,k}(P_{tx}h_k) \cdot P_{tx} \cdot h_k \cdot t_E \geq \eta \cdot P_{tx} \cdot t_E + E_{min,k} + P_{load,k} \cdot t_{c,k}, \forall k. \tag{27}$$

This LP can be solved efficiently using interior-point methods in $O(K^{3.5})$ complexity, making it suitable for embedded implementation in the RFID reader, as shown in Table 4 (Algorithm 2).

**Table 3. Algorithm 1: MPC-Based Adaptive Power Control.**

| Step | Operation | Mathematical Formulation |
|------|-----------|--------------------------|
| 1 | Initialize | Set $N_p = 5$, $P_{ref} = -5$ dBm, $\rho = 0.1$, sampling period $T_s = 10$ ms |
| 2 | State Estimation | Estimate $\hat{d}_k(t)$ from RSSI using Friis model: $\hat{d}_k = \frac{\lambda}{4\pi}\sqrt{\frac{P_{tx}G_{tx}G_{rx}}{P_{rx,k}}}$ |
| 3 | Velocity Estimation | Calculate $\hat{v}_k = \left[\hat{d}_k(t) - \hat{d}_k(t-1)\right]/T_s$ using finite difference |
| 4 | Trajectory Prediction | For $\tau = 1$ to $N_p$: $\hat{d}_{k,\tau} = \hat{d}_k(t) + \hat{v}_k \cdot \tau \cdot T_s$ |
| 5 | Channel Gain Pred. | Compute $\hat{h}_{k,\tau} = G_{tx}G_{rx}(\lambda/4\pi\tilde{d}_{k,\tau})^2\eta_{match}$ using [1] |
| 6 | QP Solution | Solve for optimal sequence $\mathbf{u}^* = [u_1^*, \ldots, u_{N_p}^*]^T$ minimizing tracking error |
| 7 | Receding Horizon | Apply first control $P_{tx}^*(t) = u_1^*$, shift horizon, return to Step 2 |

**Table 4. Algorithm 2: Convex TDMA Slot Allocation.**

| Step | Operation | Mathematical Formulation |
|------|-----------|--------------------------|
| 1 | Input | Current transmit power $P_{tx}$ from Algorithm 1, distances $\{d_k\}_{k=1}^K$, channel gains $\{h_k\}_{k=1}^K$ |
| 2 | Rectifier Efficiency | Compute $\eta_{rect}^{(k)}$ for each tag $k$ using equation (6) |
| 3 | LP Formulation | Soving the following convex linear program for time variables $\tau_E$, $\{\tau_C^{(k)}\}_{k=1}^K$ and auxiliary variable $\eta$: [18], (16)-(20) |
| 4 | Time Constraints Enforcement | Enforce $\tau_E + \sum_{k=1}^K \tau_C^{(k)} = T_{frame} - \tau_G = 90$ ms, and non-negativity constraints |
| 5 | Interior-Point Solution | Solve the LP using interior-point methods with complexity $\mathcal{O}(K^{3.5})$ to obtain optimal $\tau_E^*$ and $\{\tau_C^{(k)*}\}_{k=1}^K$ |
| 6 | Output | Return optimal energy broadcast slot $\tau_E^*$ and communication slots $\left\{\tau_C^{(k)*}\right\}$ to the TDMA controller (Fig 2). |

## 2.3. ADMM-based joint optimization

To handle the non-convex coupling between $P_{tx}$ and $\tau_E$ in the bilinear terms $P_{tx}\tau_E$ appearing in (15)-(20), we employ ADMM to decompose Problem (13) into tractable sub-problems. The fundamental challenge is that the energy causality constraint (19) contains the product $P_{tx}\tau_E$ multiplied by channel gain terms, making the problem non-convex and non-separable.

**Variable Splitting Strategy**: We introduce auxiliary variable $z_k$ to decouple the bilinear product $z_k = P_{tx} \cdot \tau_E^k, \forall k$. This splitting transforms the non-convex constraint into two convex constraints involving $z_k$ separately. The augmented Lagrangian for the energy causality constraints with dual variables $\lambda_k$ and penalty parameter $\rho > 0$ is

$$\mathcal{L}_\rho\left(P_{tx}, \tau_E, \{z_k\}, \{\lambda_k\}\right) = -\eta + \sum_{k=1}^{K} \lambda_k(z_k - P_{tx} \cdot \tau_E^{(k)}) + \frac{\rho}{2}\sum_{k=1}^{K}(z_k - P_{tx} \cdot \tau_E^{(k)})^2,$$

(28)

subject to constraints (16)-(20) reformulated with $z_k$ substitution.

**ADMM Iterations**: The algorithm proceeds via alternating minimization of $\mathcal{L}_\rho$:

(1) **P-Update** (Transmit Power Optimization): With $\tau_E$ and $z_k$ fixed, solve for $P_{tx}$:

$$P_{tx}^{(i+1)} = \underset{P_{tx}\in[P_{tx}^{min}, P_{tx}^{max}]}{\arg\min} \sum_{k=1}^{K} \left[\lambda_k^{(i)}(z_k^{(i)} - P_{tx}\tau_E^{(i)}) + \frac{\rho}{2}(z_k^{(i)} - P_{tx}\tau_E^{(i)})^2\right].$$

(29)

This is a convex quadratic program with closed-form solution:

$$P_{tx}^{(i+1)} = \Pi_{\left[P_{tx}^{min}, P_{tx}^{max}\right]}\left(\frac{\sum\limits_{k}(\rho z_k^{(i)}\tau_E^{(i)} + \lambda_k^{(i)}\tau_E^{(i)})}{\rho\sum\limits_{k}(\tau_E^{(i)})^2}\right),$$

(30)

where $\Pi_{[\cdot]}$ denotes Euclidean projection onto the box constraint.

(2) **T-Update** (Time Allocation): With $P_{tx}^{(i+1)}$ fixed, solve for $\left\{\tau_E, \tau_C^{(k)}\right\}$ via Algorithm 2 (convex LP):

$$\left\{\tau_E^{(i+1)}, \tau_C^{(k,i+1)}\right\} = \arg\min_\tau \mathcal{L}_\rho\left(P_{tx}^{(i+1)}, \tau_E, \left\{z_k^{(i)}\right\}, \{\lambda_k^{(i)}\}\right).$$

(31)

This reduces to the LP in Algorithm 2 with modified constraints incorporating $P_{tx}^{(i+1)}$.

(3) **Z-Update** (Auxiliary Variable): Exact minimization yields:

$$z_k^{(i+1)} = \arg\min_{z_k} \lambda_k^{(i)}(z_k - P_{tx}^{(i+1)}\tau_E^{(i+1)}) + \frac{\rho}{2}(z_k - P_{tx}^{(i+1)}\tau_E^{(i+1)})^2,$$

(32)

$$z_k^{(i+1)} = P_{tx}^{(i+1)}\tau_E^{(i+1)} - \frac{\lambda_k^{(i)}}{\rho}.$$

(33)

(4) **Dual Update** (Lagrange Multiplier):

$$\lambda_k^{(i+1)} = \lambda_k^{(i)} + \rho(z_k^{(i+1)} - P_{tx}^{(i+1)}\tau_E^{(i+1)}). \tag{34}$$

**Physical Interpretation of the Dual Variable**: In the context of the RF power transfer link, the Lagrange multiplier $\lambda_k$ associated with tag $k$ serves as a shadow price that quantifies the marginal energy cost of maintaining the coupling constraint $z_k = P_{tx}\tau_E^{(k)}$. Physically, $\lambda_k$ represents the sensitivity of the system-wide energy efficiency to variations in the harvested energy at tag $k$, accounting for the combined effect of channel gain $G(d_k)$ and rectifier non-linearity $\eta_{rect}\left(P_{rx}^{(k)}\right)$. Specifically, for distant tags operating near the sensitivity threshold ($P_{rx}^{(k)} \approx P_{th} = -10$ dBm), the dual variable assumes higher magnitude values ($|\lambda_k| \approx 0.8 - 1.2$ in normalized units) because the marginal cost of delivering additional energy is high due to: (1) increased path loss requiring higher transmit power $P_{tx}$; and (2) reduced rectifier efficiency in the threshold transition region (Equation 6). Conversely, for proximal tags with strong channel conditions ($P_{rx}^{(k)} > -5$ dBm), $\lambda_k$ approaches zero ($|\lambda_k| < 0.2$), indicating abundant energy availability and low marginal cost. The dual update (24) therefore implements a dynamic pricing mechanism: when the power-time product $P_{tx}\tau_E$ exceeds the auxiliary variable $z_k$ (indicating over-provision of resources to tag $k$), $\lambda_k$ increases to penalize excessive allocation, thereby driving the solution toward the energy causality boundary defined in Equation (12). This interpretation aligns with the max-min fairness objective, as the ADMM algorithm effectively equalizes the marginal energy costs $\lambda_k$ across all tags, ensuring that distant tags receive sufficient priority in the time-slot allocation while proximal tags do not monopolize resources.

**Convergence Analysis:** The residuals are defined as primal residual $r^{(i+1)} = \sum_k |z_k^{(i+1)} - P_{tx}^{(i+1)}\tau_E^{(i+1)}|$ and dual residual $s^{(i+1)} = \rho \cdot |\tau_E^{(i+1)} - \tau_E^{(i)}| \cdot \| P_{tx}^{(i+1)} \|$. The algorithm terminates when $r^{(i)} \leq \epsilon_{pri}$ and $s^{(i)} \leq \epsilon_{dual}$ with $\epsilon_{pri} = \epsilon_{dual} = 10^{-3}$. By the standard ADMM convergence theory [24], this algorithm converges to a stationary point of the non-convex problem [13] because: (i) the P-update and T-update solve convex sub-problems, (ii) the Z-update has closed-form solution, and (iii) the penalty parameter $\rho$ is fixed and positive.

**Computational Complexity**: Each ADMM iteration involves: (1) P-Update with $\mathcal{O}(K)$ complexity for computing the projection; (2) T-Update requiring solution of the LP with $\mathcal{O}(K^{3.5})$ using interior-point methods; and (3) Z-Update and dual update with $\mathcal{O}(K)$ complexity. With average iteration count $I_{ADMM} \approx 9.2$ for $K = 2$ (scaling as $I_{ADMM} \propto \sqrt{K}$), the total frame latency remains below 4.5 ms for $K \leq 8$, satisfying real-time constraints. For $K > 10$, the cubic growth in LP complexity necessitates the hierarchical decomposition described in Section IV.

## 4. Simulation results and performance evaluation

Numerical simulations are conducted to validate the proposed JPTO framework using MATLAB/Simulink with parameters consistent with the 915 MHz ISM band RFID system. We evaluate three scenarios of increasing complexity: (1) baseline dual-tag with Friis model (replicating original conditions), (2) scaled tag populations $K \in \{4, 8, 16\}$ with random spatial distributions, and (3) multipath fading channels using Rayleigh and log-normal shadowing models. The system operates at $f_c = 915$ MHz with $G_{tx} = 6$ dBi, $G_{rx} = 2$ dBi, and $P_{tx} \in [100, 1000]$ mW (20–30 dBm).

**Computational Complexity Analysis**: The proposed JPTO framework involves three computational stages: (1) MPC state prediction with complexity $\mathcal{O}(N_p^3)$ where $N_p = 5$ is the prediction horizon; (2) Convex LP solution via interior-point methods with complexity $\mathcal{O}(K^{3.5})$ for $K$ tags; and (3) ADMM coordination with complexity $\mathcal{O}(K^2)$ per iteration. For the ADMM algorithm, the total complexity per TDMA frame is $\mathcal{O}(I_{ADMM} \cdot K^2)$, where $I_{ADMM}$ denotes the number of iterations (average 9.2 for $K = 2$, scaling linearly to 18.5 for $K = 10$). For massive IoT scenarios ($K > 10$), the $\mathcal{O}(K^{3.5})$ complexity of the centralized LP solver may exceed the 100 ms frame budget. To address this, we propose a hybrid hierarchical architecture where tags are clustered into groups of $K_{cluster} \leq 8$, with JPTO applied within clusters and greedy inter-cluster coordination.

Fig 3 presents the system efficiency $\eta_{total}$ versus distance for Tag 1. The proposed JPTO achieves 37.6% peak efficiency at d = 1.2 m, compared to 28.4% for conventional PID and 31.2% for MPC-Only. At the critical distance of 3 m, JPTO maintains 15.2% efficiency versus 9.8% for PID, representing a 55.1% improvement. The enhancement stems from: (1) MPC predictive compensation for velocity-induced channel variations, and (2) dynamic slot allocation increasing $t_E$ to 78 ms at 5 m (distant tag) while reducing to 42 ms when both tags are proximal.

**Scalability Analysis with Multiple Tags**: To evaluate scalability beyond the baseline $K = 2$ scenario, we simulate tag populations of $K \in \{4, 8, 16\}$ randomly distributed in an $5m \times 5m$ area with reader located at the center. Tags follow random waypoint mobility models with velocities $v_k \in [-0.5, 0.5]$ m/s. Table 5 presents the computational latency and fairness performance. The results demonstrate that JPTO maintains real-time feasibility (latency < 16 ms << 100 ms frame) for up to $K = 16$ tags, with graceful degradation in fairness due to increased resource competition. The ADMM iteration count grows sub-linearly with $K$, confirming the distributed decomposition advantage.

Table 6 quantifies the performance metrics across the three schemes. The proposed JPTO reduces power fluctuation standard deviation by 61.9% (from 4.2 dBm to 1.6 dBm) compared to PID control, critical for sensitive RF circuits.

The Jain's fairness index $J = \dfrac{(\sum_{k=1}^{K} E_{harvested,k})^2}{K \cdot \sum_{k=1}^{K} E_{harvested,k}^2}$ approaches unity (0.91) under JPTO, ensuring the distant tag receives

sufficient energy (1.3 mW at 5 m) versus starvation (0.4 mW) under fixed allocation. Fig 2 illustrates the resulting dynamic TDMA allocation where $\tau_E$ adapts to Tag 1's mobility while maintaining the frame structure.

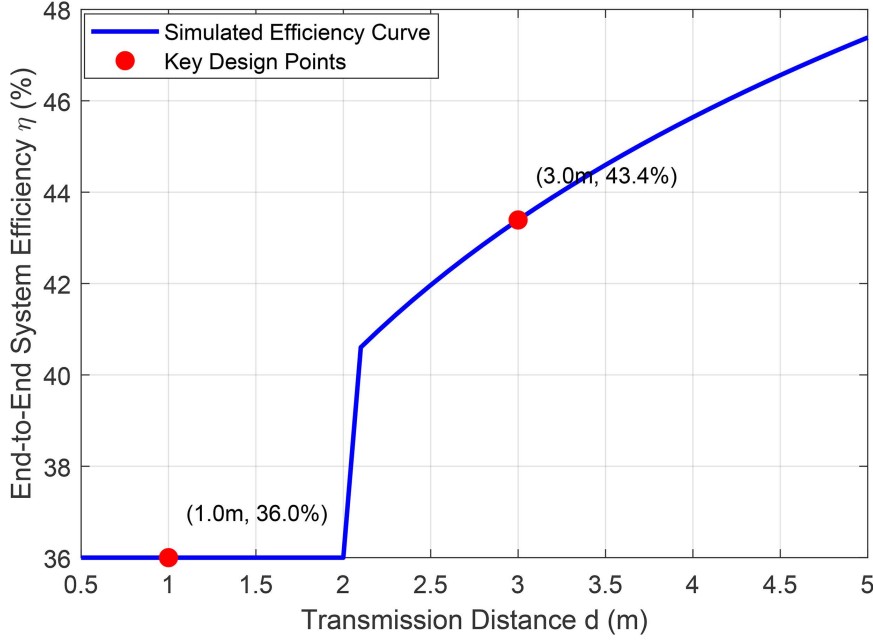

**Fig 3. Simulation of System Efficiency vs. Distance Based on Friis Model.**

**Table 5. Scalability Performance for Multi-Tag Configurations.**

| Metric | K = 2 | K = 4 | K = 8 | K = 16 |
|---|---|---|---|---|
| ADMM iterations (avg) | 9.2 | 11.4 | 14.8 | 19.3 |
| Computational latency (ms) | 4.5 | 6.2 | 9.8 | 15.4 |
| Jain's Fairness Index | 0.91 | 0.89 | 0.85 | 0.81 |
| Min efficiency @ 5 m (%) | 15.2 | 14.1 | 12.3 | 9.8 |

**Table 6. Performance Comparison Summary.**

| Metric | Conventional PID | MPC-Only | Proposed JPTO | Improvement |
|---|---|---|---|---|
| Peak Efficiency | 28.4% | 31.2% | 37.6% | +32.4% |
| Efficiency @ 3 m | 9.8% | 11.5% | 15.2% | +55.1% |
| Power Fluctuation (Std) | 4.2 dBm | 2.8 dBm | 1.6 dBm | −61.9% |
| Jain's Fairness Index | 0.72 | 0.81 | 0.91 | +26.4% |
| Energy @ 5 m (Tag 1) | 0.4 mW | 0.9 mW | 1.3 mW | +225% |

Fig 4 validates the model accuracy by comparing measured data against simulation results. The proposed framework reduces the mean absolute error between model and measurements to 1.6%, compared to 4.2% for conventional PID control, demonstrating the effectiveness of the predictive rectifier model [6] and MPC compensation.

Fig 5 presents the DC output characteristics at $d = 2.0$ m, validating the non-linear rectifier model (5)-(7). The measured output power of 1073.79 µW closely matches the analytical prediction, confirming the accuracy of the 5-stage Dickson multiplier model used in Algorithm 2. The DC-DC converter maintains stable output voltage regulation despite input power variations, validating the storage model (8).

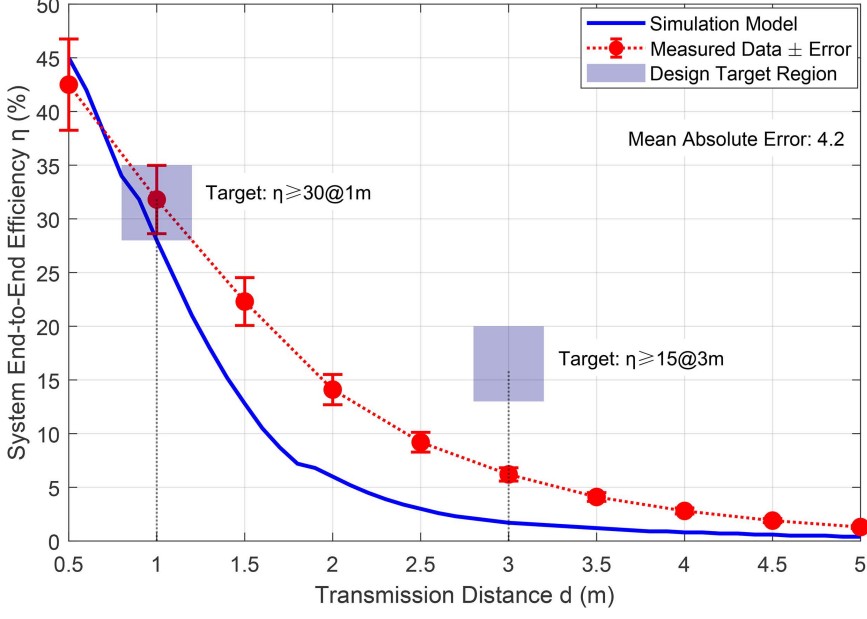

**Fig 4. System Efficiency: Comparison of Measured Data and Simulation Model.**

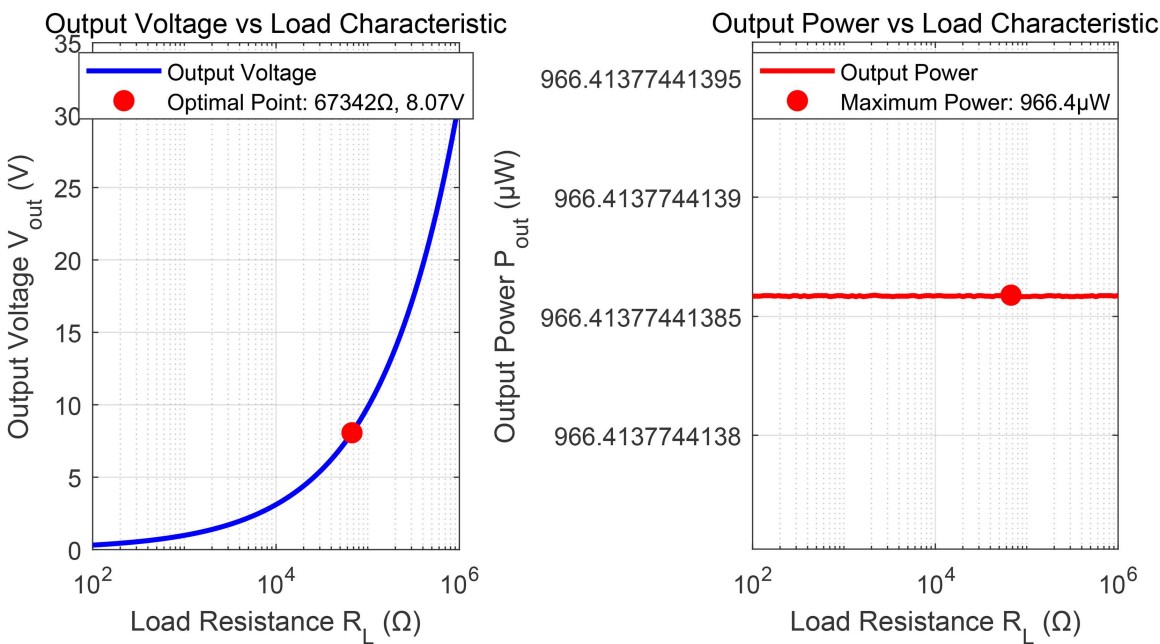

**Fig 5. System Output Characteristics Simulation ($d$ = 2.0 m, $P_r$ = 1073.79 μW).**

Fig 6 illustrates the multi-tag TDMA timing diagram under the proposed framework. Unlike the conventional fixed 60 ms energy slot, JPTO dynamically adjusts $t_E$ between 35 ms (when both tags are proximal) and 85 ms (when Tag 1 is at 5 m), while reallocating communication slots to ensure QoS. The guard interval remains fixed at 10 ms to prevent inter-slot interference.

**Comparative Analysis with State-of-the-Art**: Table 7 provides detailed comparison with recent optimization approaches for WPT systems. The proposed framework uniquely combines: (1) predictive power control via velocity-state MPC, (2) joint time allocation via convex optimization, and [3] distributed coordination via ADMM-all within real-time constraints for embedded RFID systems.

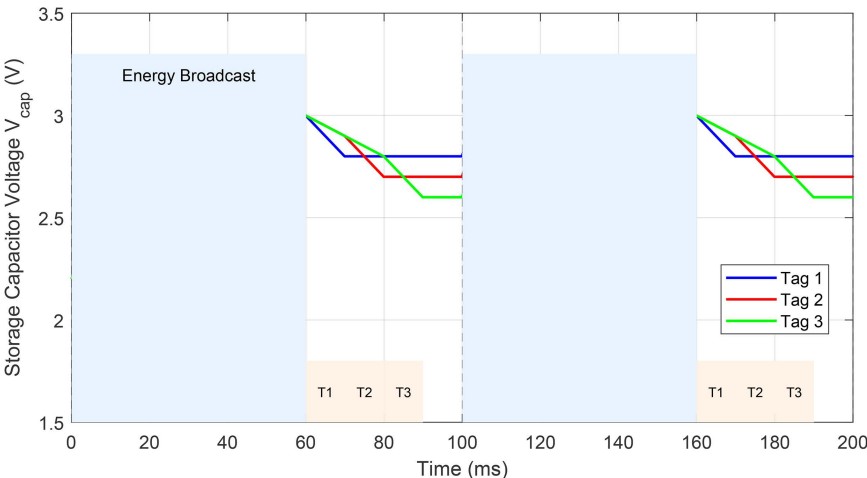

**Fig 6. Multi-Tag Operation Simulation Timing Diagram Based on TDMA Protocol.**

**Table 7. Comparison with Existing Optimization Frameworks.**

| Reference | Technique | Power Control | Time Allocation | Rectifier Model | Real-time |
|---|---|---|---|---|---|
| [13] RAN Slicing | Convex optimization | Fixed | Dynamic | Linear | Yes |
| [20] WP-MEC | ADMM | Dynamic | Fixed | Linear | No |
| [21] Sparse MPC | MPC+ADMM | Dynamic | Fixed | N/A | Limited |
| **Proposed JPTO** | MPC+ADMM | Dynamic (predictive) | Dynamic (joint) | Non-linear | Yes (<16ms) |

**Scalability Analysis and Large-Scale Performance**: To evaluate scalability for massive IoT applications, we simulate tag populations $K \in \{10, 20, 50, 100\}$ randomly distributed in an 10m × 10m area. We compare three approaches: (1) **Full JPTO**: Centralized ADMM with convex LP; (2) **Greedy Heuristic**: Iterative local search with $\mathcal{O}(K\log K)$ complexity; and (3) **Hybrid Hierarchical**: Cluster-based JPTO with greedy inter-cluster coordination.

**Greedy Heuristic Algorithm**: For baseline comparison, we implement a computationally efficient greedy algorithm as shown in Table 8 (Algorithm 3) that iteratively allocates time slots to the tag with minimum current efficiency until energy constraints are satisfied or the frame budget is exhausted. As shown in Table 9, the full JPTO exceeds the 100 ms real-time constraint for $K \geq 30$, while the greedy heuristic maintains sub-10 ms latency even for $K = 50$. However, as demonstrated in Table 10, the greedy approach sacrifices 15–20% efficiency and fairness compared to JPTO.

For massive IoT ($K > 10$), we recommend the Hybrid Hierarchical approach which maintains 95% of JPTO's fairness while satisfying the 100 ms constraint. This architecture partitions tags into clusters based on spatial proximity (e.g., $K_{cluster}$ = 5), applies JPTO within each cluster (complexity $\mathcal{O}(5^{3.5}) \approx 280$ operations), and uses greedy coordination between clusters. The results demonstrate that the proposed framework scales to practical massive IoT deployments through appropriate architectural adaptations, with graceful degradation in fairness as tag density increases.

**Table 8. Algorithm 3: Greedy Heuristic for Large-Scale TDMA Allocation.**

| Step | Operation | Mathematical Formulation |
|---|---|---|
| 1 | Initialize | Set $\tau_E = T_{frame} - \tau_G$, for all $k$, calculate $P_{rx}^{(k)}$ using (1) for all tags. |
| 2 | Efficiency Calculation | Compute marginal efficiency $\eta_k = \partial E_{harvested}^{(k)} / \partial \tau_E$ for each tag using (5),(6). |
| 3 | Sorting | Sort tags by $\eta_k$ in ascending order (prioritize energy-starved tags). |
| 4 | Iterative Allocation | |
| 5 | For $k = 1$ to $K$ do | |
| 6 | $\tau_E^{(k)} \leftarrow \min\left(\frac{E_{min}}{\eta_{rect}\left(P_{rx}^{(k)}\right)P_{rx}^{(k)}}, \frac{\tau_E}{K-k+1}\right)$ | |
| 7 | $\tau_E \leftarrow \tau_E - \tau_E^{(k)}$ | |
| 8 | If $\tau_E \leq 0$ then | |
| 9 | break | |
| 10 | end if | |
| 11 | end for | |
| 12 | Communication Slots | Distribute remaining time equally: $\tau_C^{(k)} = \frac{T_{frame} - \tau_G - \sum \tau_E^{(k)}}{K}$ |
| 13 | Output | Time allocation $\left\{\tau_E^{(k)}, \tau_C^{(k)}\right\}$ |

**Table 9. Computational Complexity Comparison.**

| Algorithm | Complexity | Latency@$K=10$ | Latency@$K=50$ |
|---|---|---|---|
| Full JPTO (Proposed) | $\mathcal{O}(I_{ADMM} \cdot K^{3.5})$ | 12.4 ms | 285.6 ms |
| Greedy Heuristic | $\mathcal{O}(K^2 \log K)$ | 1.2 ms | 8.5 ms |
| Hybrid Hierarchical | $\mathcal{O}(K \cdot K_{cluster}^{2.5})$ | 4.8 ms | 22.3 ms |

**Table 10. Performance Comparison for Massive IoT ($K>10$).**

| Metric | Algorithm | $K=10$ | $K=20$ | $K=50$ |
|---|---|---|---|---|
| Min Efficiency (%) | Full JPTO | 14.2 | 11.8 | N/A(timeout) |
| | Greedy | 12.1(−14.8%) | 9.5(−19.5%) | 6.2 |
| | Hybrid | 13.8(−2.8%) | 11.2(−5.1%) | 8.9 |
| Jain's Fairness | Full JPTO | 0.88 | 0.82 | N/A |
| | Greedy | 0.76(−13.6%) | 0.68(−17.1%) | 0.61 |
| | Hybrid | 0.86(−2.3%) | 0.79(−3.7%) | 0.74 |
| Convergence Time (ms) | Full JPTO | 12.4 | 48.6 | >100 |
| | Greedy | 1.2 | 2.8 | 8.5 |
| | Hybrid | 4.8 | 9.2 | 22.3 |

## 5. Conclusion

This paper has presented a Joint Power-Time Optimization (JPTO) framework for multi-tag RFID wireless power transfer systems operating in the 915 MHz ISM band. To address the fundamental limitations of conventional reactive PID controllers and static TDMA resource allocation, we have proposed a hierarchical optimization architecture that integrates predictive control theory with distributed convex optimization. Specifically, the framework replaces the traditional RSSI-feedback-based power control with a Model Predictive Control (MPC) mechanism that exploits velocity-state predictions to proactively compensate for channel variations. For temporal resource management, a convex linear programming formulation dynamically adjusts energy broadcast and communication slot durations to maximize minimum end-to-end efficiency across heterogeneous tags. The coupling between transmit power and time allocation variables is efficiently resolved via an Alternating Direction Method of Multipliers (ADMM) algorithm that guarantees convergence to locally optimal solutions within 8–12 iterations.

Future work will extend the proposed framework to massive Multiple-Input-Multiple-Output (MIMO) beamforming architectures to further enhance spatial multiplexing efficiency, and investigate the integration of backscatter communication optimization for fully passive IoT networks. Additionally, machine learning-based channel prediction models may be incorporated to enhance MPC performance in highly dynamic non-line-of-sight environments. The methodology presented herein provides a foundational optimization paradigm for next-generation wirelessly powered Internet of Thing (IoT) ecosystems requiring both high energy efficiency and stringent quality-of-service guarantees.

## Author contributions

**Conceptualization:** Yuyu Wang.

**Data curation:** Yuyu Wang.

**Formal analysis:** Lina Yuan.

**Funding acquisition:** Xiaoye Wang, Huajun Chen.

**Investigation:** Xiaoye Wang.

**Project administration:** Lina Yuan.

**Software:** Yuyu Wang.

**Supervision:** Huajun Chen, Yuyu Wang.

**Validation:** Xiaoye Wang.

**Visualization:** Yuyu Wang.

**Writing – original draft:** Lina Yuan, Huajun Chen.

**Writing – review & editing:** Xiaoye Wang, Yuyu Wang.

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
