## [Decision Letter · Decision Letter 0]

23 Mar 2026

Dear Dr. Chen,

Thank you for submitting your manuscript to PLOS ONE. After careful consideration, we feel that it has merit but does not fully meet PLOS ONE’s publication criteria as it currently stands. Therefore, we invite you to submit a revised version of the manuscript that addresses the points raised during the review process.

We look forward to receiving your revised manuscript.

Kind regards,

Zhiyuan Ren

Academic Editor

PLOS One

Journal Requirements:

“This research was supported by 2026 Annual Doctoral Research Initiation Fund Project of Tongren University (trxyDH2622), and in part by Guangdong Basic and Applied Basic Research Foundation (Grant No. 2024A1515140010).”

“This research was supported by 2026 Annual Doctoral Research Initiation Fund Project of Tongren University (trxyDH2622), and in part by Guangdong Basic and Applied Basic Research Foundation (Grant No. 2024A1515140010).”

“This research was supported by 2026 Annual Doctoral Research Initiation Fund Project of Tongren University (trxyDH2622), and in part by Guangdong Basic and Applied Basic Research Foundation (Grant No. 2024A1515140010).”

5. In the online submission form you indicate that your data is not available for proprietary reasons and have provided a contact point for accessing this data. Please note that your current contact point is a co-author on this manuscript. According to our Data Policy, the contact point must not be an author on the manuscript and must be an institutional contact, ideally not an individual. Please revise your data statement to a non-author institutional point of contact, such as a data access or ethics committee, and send this to us via return email. Please also include contact information for the third party organization, and please include the full citation of where the data can be found.

Reviewers' comments:

Reviewer's Responses to Questions

**Comments to the Author**

1. Is the manuscript technically sound, and do the data support the conclusions?

Reviewer #1: Yes

Reviewer #2: Yes

2. Has the statistical analysis been performed appropriately and rigorously?

Reviewer #1: Yes

Reviewer #2: Yes

3. Have the authors made all data underlying the findings in their manuscript fully available?

Reviewer #1: Yes

Reviewer #2: Yes

4. Is the manuscript presented in an intelligible fashion and written in standard English?

Reviewer #1: Yes

Reviewer #2: Yes

Reviewer #1: The paper has several positive aspects. First, it addresses an important challenge in RFID-based wireless power transfer systems: the inefficiency of static TDMA allocation and reactive PID-based power control. By introducing predictive power control using MPC and adaptive time allocation, the framework aims to improve both energy efficiency and fairness among tags located at different distances from the reader. Second, the hierarchical optimization structure (MPC for power, linear programming for time allocation, and ADMM for coordination) is technically reasonable and potentially suitable for embedded implementation. Third, the study considers practical system constraints such as transmit power limits, rectifier efficiency, capacitor storage capacity, and guard intervals, which increases the realism of the model. Simulation results show improvements in efficiency, fairness, and power stability compared with conventional PID control and MPC-only schemes.

Despite these strengths, several issues limit the current quality of the paper. The most significant concern is that the novelty of the work is not sufficiently demonstrated. Many of the techniques used such as MPC, convex optimization for TDMA allocation, and ADMM for decomposing non-convex problems are already widely used in optimization and wireless communication research. The manuscript does not clearly explain how the proposed framework differs from existing studies or what specific methodological contribution it introduces. A stronger comparison with related literature is necessary.

Additionally, the problem formulation lacks clarity. Several equations are difficult to interpret due to inconsistent notation and incomplete definitions of variables, particularly the exact expression for the energy efficiency objective function. The ADMM formulation also appears insufficiently justified, and the derivation of the update steps should be explained more rigorously.

Another limitation is the restricted simulation setup, which only considers two RFID tags and a simplified Friis free-space channel model. Real RFID deployments often involve many tags and more complex propagation conditions such as multipath fading or shadowing. Evaluating the proposed framework under larger tag populations and more realistic channel models would strengthen the validity of the results.

In conclusion, the paper addresses a relevant problem and proposes a potentially useful optimization framework, but improvements are needed in novelty justification, theoretical clarity, and experimental evaluation. With clearer formulations, stronger comparisons with existing work, and more comprehensive simulations, the manuscript could make a meaningful contribution to RFID-based wireless power transfer research.

Reviewer #2: The paper proposes a Joint Power Time Optimization framework designed to enhance energy harvesting efficiency and fairness for mobile IoT tags in RFID networks. The system manages the nonlinearities of the wireless power link. I have the following comments:

1. The rectifier efficiency is modeled using a tanh function in (6). Please provide a comparison between this analytical approximation and the actual physical behavior of the 5 stage Dickson multiplier at the sensitivity threshold go -10 dBm.

2. As the number of tags increases for massive IoT applications, this complexity will quickly consume the available time budget. Discuss the scalability of this approach and order a comparative analysis with a greedy heuristic for $K > 10$

3. In the introduction, some ISAC works can be included besides [5] such as https://doi.org/10.1109/TWC.2025.3645048 and https://doi.org/10.1109/TWC.2024.3357573

4. In the ADMM formulation, the dual variable coordinates the solutions between the power update and the time update. Provide a physical interpretation of this variable in the context of the RF power transfer link. Is it representative of the marginal energy cost for the distant tag?

5. Kindly compare JPTO against a fixed power fixed time allocation baseline to quantify total energy gain.

6. Please check if the ADMM algorithm with $K=50$ tags to measure the actual computational latency.

7. It would be good to simulate a case where you have a tag movement mobility with a Gauss Markov mobility model to test the MPC prediction robustness.

8. The Jain's fairness index is calculated based on harvested energy. However, different tags may have different $P_{load}$ requirements based on their specific sensor types. A more accurate simulation would measure fairness based on the ratio of harvested energy to required energy for each tag.

9. Please fix the formatting of the paper. For instance, page 9 has many whitespaces in the contribution part and Figure 1 has small text.

.

Reviewer #1: No

Reviewer #2: No

---

## [Author Response · Author response to Decision Letter 1]

6 Apr 2026

Please see the Attach File-Respose to Reviewers!

---

## [Decision Letter · Decision Letter 1]

8 Apr 2026

Joint Power-Time Resource Optimization for Multi-Tag RFID Wireless Power Transfer via Model Predictive Control and ADMM

PONE-D-26-05818R1

Dear Dr. Chen,

We’re pleased to inform you that your manuscript has been judged scientifically suitable for publication and will be formally accepted for publication once it meets all outstanding technical requirements.

Kind regards,

Zhiyuan Ren

Academic Editor

PLOS One

Additional Editor Comments (optional):

Reviewers' comments:

Reviewer's Responses to Questions

**Comments to the Author**

Reviewer #1: All comments have been addressed

Reviewer #2: (No Response)

2. Is the manuscript technically sound, and do the data support the conclusions?

Reviewer #1: Yes

Reviewer #2: (No Response)

3. Has the statistical analysis been performed appropriately and rigorously?

Reviewer #1: Yes

Reviewer #2: (No Response)

4. Have the authors made all data underlying the findings in their manuscript fully available?

Reviewer #1: Yes

Reviewer #2: (No Response)

5. Is the manuscript presented in an intelligible fashion and written in standard English?

Reviewer #1: Yes

Reviewer #2: (No Response)

Reviewer #1: (No Response)

Reviewer #2: I have the following minor comments:

1. The introduction can include more ISAC works such as [REF01] and [REF02].

2. Please double check the consistency of the citations.

3. The titles, axis labels and legends for the figures are too small to be legible (e.g. Fig. 4). Please enlarge the font size for all the figures.

References

[REF01] A. Chowdary, et al. "On Hybrid Radar Fusion for Integrated Sensing and Communication," in IEEE Transactions on Wireless Communications, vol. 23, no. 8, pp. 8984-9000, Aug. 2024, doi: 10.1109/TWC.2024.3357573.

[REF02] E. Illi, et al. "On the Secrecy-Sensing Optimization of RIS-Assisted Full-Duplex Integrated Sensing and Communication Network," in IEEE Transactions on Wireless Communications, vol. 25, pp. 9530-9547, 2026, doi: 10.1109/TWC.2025.3645048.

.

Reviewer #1: No

Reviewer #2: No

---

## [Editor Report · Acceptance letter]

PONE-D-26-05818R1

PLOS One

Dear Dr. Chen,

I'm pleased to inform you that your manuscript has been deemed suitable for publication in PLOS One. Congratulations! Your manuscript is now being handed over to our production team.

Kind regards,

on behalf of

Professor Zhiyuan Ren

Academic Editor

PLOS One